# A Conversation about Ethics: A Deliberative and Practice-Based Approach to Ethics in Arts Education

**Samantha Broadhead** [1],*[ID]**, Karen Tobias-Green** [2] **and Sharon Hooper** [3]

1   Research, Leeds Arts University, Leeds LS2 9AQ, UK
2   Creative Writing, Leeds Arts University, Leeds LS2 9AQ, UK
3   Visual Communication, Leeds Arts University, Leeds LS2 9AQ, UK
*   Correspondence: sam.broadhead@leeds-art.ac.uk

**Abstract:** This article reports on a practice-based research project that examined the various orientations of practice to ethical deliberation. The aim was to produce a film that captured ethical debate between two creative practitioners as they walked through their local streets. The film would be a catalyst for staff and students at an arts institution to think about their own ethical practices. The approach taken was based on Aristotelian notions of phronesis or practical wisdom, which is concerned with making ethical judgments based on deliberation. Issues were raised by the project, such as the tensions between policy and practice and the tensions between aesthetic considerations and ethical practice. Questions about the value of narrative, representation, and learning through doing were raised by the work.

**Keywords:** practice-based research; digital video; film; arts; ethics; ethical practice; ethics policy; phronesis; Aristotle

## 1. Introduction and Context

Historically, explicit discourses about ethics within an art school may not have been prevalent. However, as institutions evolve in accordance with the changing contexts of higher education, their staff and students are required to grapple with research codes of practice and ethical frameworks [1–3]. Many of these frameworks are informed by processes and points of reference borrowed from multi-faculty universities. As a result, ethics is seen through the lenses of scientists or sociologists. In the U.K., attempts to establish national standards for research ethics support through the United Kingdom Research Integrity Office (UKRIO) appear to focus on models more aligned to the sciences than the arts and humanities [4]. Academics from creative subject areas need to make sense of ideas that may at first seem alien and inappropriate for those working with artistic ways of knowing [5]. Fears may arise that ethical concerns may constrain innovation and creative agency.

There have been practice-based and arts-based researchers who have thought deeply about ethical practice. Hübner (2021) argued that ethics is central to all research including practice-based research, and that ethics policies offer general guidance but much is left to the judgement and integrity of the researcher [6]. O'Donoghue (2009) considered that researching in and with the arts requires a different relationship with the practices and procedures that have come to define appropriate ethical conduct [7]. Furthermore, Barine and Eisner (2011) asked how arts-based research can be political, ethical and artful, while Finely et al. (2020) suggested that we need to connect politics, ethics and pedagogy with action [8,9].

This article considers how practiced-based research can open up debates about ethics in a specialist arts context. The project was undertaken in a small specialist arts university, which specialises in creative subjects such as art, design, film, music and creative writing. Currently, there is an enrolment of approximately three thousand students, mostly

comprised of undergraduates and a growing cohort of one hundred and twenty postgraduate students. In addition, a second campus provides further education courses in art and design.

During the previous five years the institution underwent a period of transition and change, from an art and design college, to an arts University. In 2017, the College gained University title.

Some ethical guidelines existed when previously a college, however, these were later formalised into an ethics policy for research and teaching. The model for the policy came initially from those developed by multi-faculty universities that accommodated subjects such as natural and social sciences. Through various iterations, the ethics policy became more attuned to arts practices. A sub-committee of the Research Committee was responsible for overseeing the ethical practices of research and teaching. It was the Ethics Sub-Committee that identified the need for training staff and students in ethical practice.

Three researchers (a creative writer, a documentary maker and a fine artist) took up the call from the Ethics Sub-Committee and worked on a proposal called, 'A conversation about ethics'.

The project was a collaboration between the three researchers that aimed to make a film that captured a discussion about ethics in an arts context between two people walking in the street as part of their everyday lives. This approach was taken to show how ethics was situated in life rather than something that remained in the realm of academia [10]. The theoretical underpinning for this direction ultimately came from Ancient Greek philosophy that saw moral value as inseparable from other values and human attributes [11]. Within the context of this article, the term 'moral' is related to the everyday conduct and rules of behaviour that people adhere to, rather than the religious connotation. Ethics is understood to be the philosophical reflection on morals, right action, and the good life. This approach was deemed to make ethics more meaningful and democratic. At the same time, it can be seen as being in tension with the concept of an ethics policy that could be described as a set of codes that, although were well meaning, separate ethical practices from other forms of practices in the University and life in general.

## 2. Theoretical Position

The theoretical approach was informed by Aristotle's understanding of ethics and practical wisdom [12]. Nussbaum's (2001) fragility of goodness was also a significant influence on the project, as she examined Aristotle's work and saw its relevance in contemporary societies [11]. Aristotle (1953) started his reasoning with the quest to explore how we can lead a rich, good life. He did not separate moral value from other parts of human existence. This allowed the complexity of living a good life to be investigated, including virtues, such as courage, justice or temperance, and also capacities, such as intellect and the ability to reason and make judgments [13–15].

Aristotle recognised that ethics could be not understood through a single measure, such as whether or not actions lead to happiness. Complex ethical issues cannot be judged with a 'straight edge' in a practical context [16]. Perception, openness and responsiveness are needed to respond to situations which are often non-repeatable [17,18].

Nussbaum (2001) pointed to ethical dilemmas where there are two or more conflicting outcomes of moral action that show that leading a good life is not always possible, being subject to outside influences [11]. For example, a parent needs to work in order to feed their family, but this leads to great unhappiness as the family's emotional needs are neglected. Another example from art and design education is that a teacher may wish to organise a trip because of the educational benefits for their students, however, some poorer members of the cohort who cannot afford the costs will be disadvantaged [19]. Nussbaum writes,

> 'We are able to deliberate and chose, to make a plan in which ends are ranked to decide actively what is valued and how much. . . . It seems possible that this rational element in us can rule and guide the rest, thereby saving the whole person from living at the mercy of luck.' [11]

Luck describes the impact of circumstances that are outside the control of the individual. Nussbaum continued to say there are limits to human agency, there may be circumstances where any possible choice of action will lead to undesired consequences that cause anguish and pain.

Underpinning the concept of an ethics policy is that ethical practices are possible if people are guided to act well. This position believes that rational ethical reasoning, judgment and action can also lead to the correct outcomes in any given situation.

> 'For the Kantian believes that there is one domain of moral value, that is altogether immune to the assaults of luck. No matter what happens in the world the moral value of the good will remains unaffected. Furthermore, the Kantian believes that there is a sharp distinction to be drawn between this and every other type of value, and moral value is of overwhelmingly greater importance than anything else.' [11]

In the practical context it may not always be so clear what the right thing to do is, due to the complexity and messiness of life. Emotions such as love, passion, fear and envy will also influence ethical practices; how realistic is it for people to remain objective in all situations? The Greek thinkers argued that moral value is vulnerable to luck [11,20]. Goodness is seen to be possible, but not certain; it is fragile.

The Aristotelian view of deliberation guided by phronesis or practical wisdom is an approach to ethics which tries to deal with these complexities and at the same time acknowledges the fragility of goodness; that it cannot always be facilitated by policies, rules and codes.

The person of practical wisdom needs the capacity of perception to recognise when a situation demands careful deliberation and judgment [18]. Korthagan et al. (2001) built on the work of Plato and Aristotle in order to distinguish between two types of knowledge—episteme and phronesis [17]. Episteme is defined as abstract, objective knowledge which is derived by generalising about many situations. Phronesis is the practical wisdom derived from the perception of a particular context. They argued that 'It is the eye that one develops for a typical case, based on the perception of particulars' [18].

Korthagen et al. (2001) pointed out that a person can only gain practical wisdom with much experience of perceiving, assessing situations, choosing courses of action and evaluating the consequences of action [18]. Practical wisdom cannot be gained by only learning universal laws or conceptual principles. However, these do act as guides to inform deliberations about future actions.

Phronesis considers the interplay between the particular and the universal. The aim of practical wisdom is to decide how to act ethically, in the best interests of the self and other people [16]. Such deliberations need to be cognisant of the contexts in which these decisions are made, and possible implications they may have. The question for this project is: how can people be encouraged to deliberate well about ethical practices? As Larmore (1987: 15) writes:

> Judgement itself, [Aristotle] stressed, is not an activity governed by general rules; instead, it must always respond to the peculiarities of the given situation. Thus, no one can acquire judgement by being imparted some kind of formal doctrine. It can be learned only through practice, through being trained in the performance of right actions. [21]

According to Aristotle, practical reason is gained only through active participation and involvement in public life. Therefore, people should be encouraged to practice deliberation in their day-to-day lives so they become adept at it, not relying solely on the guidance of a policy.

Alternatively, Nussbaum (in Wall, 2005) argued that reading novels, tragedies, i.e., fictional literature, provides a form of education in practical wisdom through their explorations of human dilemmas and conflicts [22]. Stories help us perceive and attend to the particularities of contexts and persons around us [20,22]. The purpose of practical wisdom

is to respond to the complexity of people's lives with 'moral perception', moral imagination and moral sensibility. As Nussbaum (1990: 184) explained, 'Stories cultivate our ability to see and care for particulars, not as representatives of a law, but as what they themselves are: to respond vigorously with senses and emotion as before. . . . To wait and float and be actively passive' [23].

Literary narratives allow people to imagine, or think, about possible consequences of their actions and judgments by reading fictions and other people's accounts [13,20,23]. The things that are learned need to be put into action before the individual can become adept at using their practical wisdom.

To summarise, the theoretical approach that underpins this project is based on an Aristotelian understanding that ethics are concerned with the universal and the particular. Universal codes are not adequate on their own to guide people in their ethical practice. Living a good life is possible but it is fragile and subject to outside influences over which the individual has little control. Practical wisdom as in a process of deliberation is one way of making good ethical decisions that are context bound and responsive to unrepeatable situations. Narratives are one way in which people can be developed so they are confident in their own deliberations.

### 3. Methodology

The method aligns most closely with practice-based research because the outcomes include reflections on practice as well as the creation of a film. The researchers aim was to gain new knowledge partly by means of practice, and the outcomes of that practice [1,24]. The research was practice-centred, flexible, process- and product-driven [24,25]. Insights gained through undertaking the project would be realised in the practical output and also through the critical reflection in and on the practice [26,27]. Considered introspection which is later shared with other practitioners is inherent in the creative process, and was utilised by the researchers in this project [28–30]. As practice-based researchers, they were 'obliged also to map for his or her peers the route by which they arrived at their product/s' [31]. This approach was chosen because, 'It is a way of acknowledging that not everything that is knowable or worth knowing can be captured accurately within mathematical or scientific frameworks or . . . theoretical orthodoxies' [25]. The practice-based approach was appropriate because, 'It allows the research work of the creative practitioner to ask questions not only . . . about work . . . but through work.' [28]. This article attempts to map the creative processes as well as make explicit the theoretical starting points and emerging discoveries.

### 4. Materials and Methods

In the spirit of the Aristotle's understanding of living a good life that is inseparable from ethical practice, the project sought to embed ethical discourses within the everyday patterns of existence. At the same time, the idea that ethical deliberation is caught within a conversation was seen to be important, because it demonstrated how ethical questions could be asked and openly explored. A film with a simple narrative was devised where two people go for a walk on a summer's day through the streets surrounding their university, they stop for a coffee in a local café, then meander through the cityscape, ending their journey in a celebrated local cemetery. During the walk, the two protagonists discuss what ethics means to them, alluding to various aspects of ethical practices such as informed consent, authorship, collaboration, respect for living creatures, and sustainable practices. Here the ethics policy was a starting point and a loose guide for the topics under discussion, however, the speakers were able to direct the conversation based on their own experiences and judgements. The approach was evocative of Pink's (2007) work as it employed a similar method, the video tour. The researcher, with the participants, walked around their homes, video recording as they 'showed', performed and discussed their habitats. Pink explained how 'walking with video' creates a form of walking sociality between the researcher, camera and the video subject [32].

As has already been mentioned, it was deemed that the film was best situated within the street. This was a place which would be very familiar to both staff and students. The stylistic references in the film derived from two main sources. Firstly, the work of Agnes Varda (1961), and more specifically, *Cleo from 5 to 7*, which is an example of French new wave cinema [33,34]. The protagonist spends two hours in real time walking through the streets of Paris, contemplating possible fears about her health and an imminent visit to her doctor. The second source was *Examined Life*, directed by Astra Taylor (2008) [35]. The film features eight important contemporary philosophers walking around New York and other metropolises, discussing the practical application of their ideas in contemporary culture.

Situating the film in the street rather than in a studio or a classroom presented many technical problems to resolve, such as capturing the dialogue clearly, creating a feeling of continuity, and moving safely through crowds of people. However, in spite of these issues, it was decided that it was important that the film captured conversations in motion and was situated in the local city environment.

A loose and adaptable 'script' was agreed upon by the three researchers. The dialogue was performed and filmed in a classroom so that the protagonists were able to practice and obtain a sense of the flow and ebb of the dialogue. On viewing the footage, it was confirmed that the project really did need the dynamism of walking in the street to come alive.

The filming took place during one day. The three researchers worked together, however the documentary maker took the lead on the technical decisions. The 'actors' did not closely follow the script, but spoke with spontaneity after deciding which topics they would broadly cover.

After filming, the process of editing was time-consuming and took a number of weeks. This was an opportunity for critical reflection, and much of the analysis occurred during this stage of the process.

The researchers met at regular intervals, refining the work so that it had a naturalistic style and continuity. Footage from one day's filming needed to be reduced to fifteen minutes. It was thought that this was just enough time to cover all the significant moments without being too long.

During this time, unanticipated ethical issues arose. The issue of filming in the street led to a discussion about informed consent. Some people who were going about their daily business had been captured on the film. The employees in the café and a customer who was sitting at a table reading had already given their permission to be in the film. However, the camera had lingered on a couple of people's faces, which had been unanticipated. This occurred during important sequences of dialogue and so could not be easily cut. At this point, the research team consulted the University ethics policy, and Channel Four (2020) and BBC (2020) guidelines, for ideas on how to solve this problem [36,37]. It was decided to blur out the faces, although this did disrupt the beauty and naturalism of the film.

When the film was finished, credits were given to the makers of the film, and also those who had helped with the filming on the day, and the later editing. *A Conversation about Ethics* was first screened to the Ethics Sub-Committee, who commented on how comprehensive it was in dealing with a range of ethical issues. They did request that the film be given subtitles. That the film needed to be accessible was another consideration that had eluded the small research team. Therefore, the film was re-edited with sub-titles [38].

The film was screened to staff and students and then made accessible through the institution's virtual learning environment and research repository. It was important that the output of the research project was made open access, therefore, the concern about identifying individuals who had not given their consent was even more valid.

## 5. Results

The method resulted in a 15 min video in which two researchers (see Figure 1). (Broadhead and Tobias-Green discussed what ethics is and why it matters. The content of the video was written/improvised and performed by Broadhead and Tobias-Green. The video was directed by the third researcher (Hooper). The video provided a platform for them to

share their thoughts on ethical research practice with one another, generating an output which provided a clear outline of ethical practice from their combined wealth of experience.

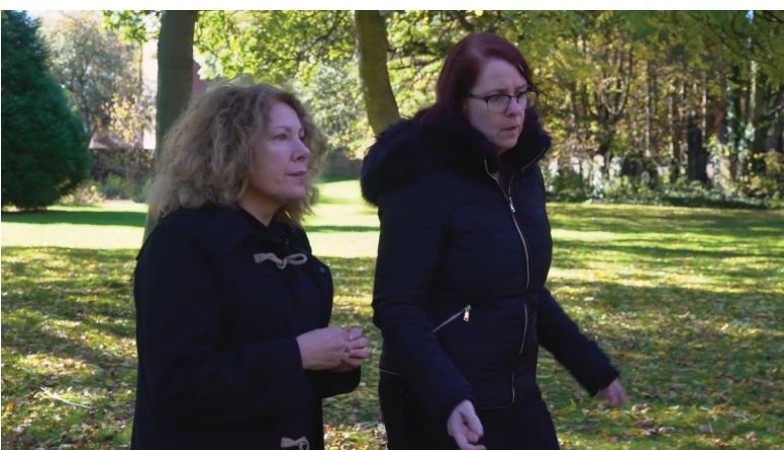

**Figure 1.** A still from *A Conversation about Ethics*, showing the discussion while walking. Directed by Sharon Hooper.

The resulting video sequences suggested to the viewer which questions they need to ask themselves when they embark on an arts research project, and the best practice to follow before collecting data or reflecting on practice to produce a creative output. The film was shared at the Media Practice Education/Media Communications and Cultural Studies Association at Solent University, South Hampton 21–25 June 2021. It can be viewed at https://lau.repository.guildhe.ac.uk/id/eprint/17574/ (accessed on 25 June 2021).

## 6. Discussion

In making the film, it was seen that ethical deliberation occurred through at least three orientations to the film: ethical deliberation represented in the film; ethical deliberation in making the film; and ethical deliberation about the film.

### 6.1. The First Orientation

The ethical deliberation represented in the film demonstrated the interplay of the universal and the particular. Aristotle commented on how universal laws need to be modified in light of particular circumstances. Laws are an inadequate guide for ethical judgments that require the responsiveness of practical wisdom. As the camera recorded the two protagonists walking and discussing ethical protocols—for example, those about informed consent—the gaze of the camera would fall upon the specific unique textures such as stone or vegetation existing in that specific place. This could be read as a metaphor for the complexity inherent in deliberations that draw upon the universal and particular.

All law is universal—but about some things it is not possible for a universal statement to be correct. Then, in those matters in which it is necessary to speak universally, but not possible to do so correctly, the law takes the usual case, though without ignoring the possibility of missing the mark. When, then, the law speaks universally, and something comes up that is not covered by the universal, then it is correct, insofar as the legislator has been deficient or gone wrong in speaking simply, to correct his omission, saying he would have said himself if he had been present and would have legislated had he known [16]. The University ethics policy is, in essence, a set of laws about good conduct in research and teaching. It comprises a summary of many previous wise decisions, but, as such, is always backward looking. The past decisions, those that underpin the guidance of the ethics policy, ultimately may have roots in scientific subjects and so do not always address aesthetic concerns. Therefore, openness and flexibility are essential in deliberating about new situations. Each new instance of an ethical deliberation should modify and improve policy. Therefore, although the conventional themes found in many ethics policies such as

informed consent and research integrity were used as starting points in the discussion, the random encounters in the street experienced by the two protagonists guided the direction of the dialogue. One critical comment about the film would be that the two protagonists were often in agreement, and maybe differences in perception and understanding would have demonstrated the deliberative process more effectively. Ethical deliberation was made visible by the film and constituted the first orientation.

*6.2. The Second Orientation*

The second orientation was the ethical deliberation that happened while making the film. This occurred throughout the planning, filming and editing sections of the film. It is possible that the subject of the project led to a self-consciousness about the reflexive processes that seemed to aid the discussion between the three researchers. There were initial discussions and then agreement about equal authorship of the research project's outcomes, followed by who would be acknowledged for their work in helping with filming and editing. Gaining informed consent from people who would be accidently in the film was not rigorously thought through at this point. During the filming, there was some reflection-in-action, when in the cemetery the two protagonists began walking on some gravestones which had been turned into a pavement. At first this was undertaken unthinkingly, but then a discussion arose about whether this was the right thing to do, that we were in effect treading on people's life stories. This realisation caused the researchers to think about the decentring of the self, the decentring of the 'rights' of the individual practitioner to say and do exactly what they wanted without reflection or regard for the invisible and other people. Braidotti, (2013) argued that new materialism shifts the human from the focus of attention, establishing ethical practice that can engage with human culture, other living things, and the environment of inanimate matter, such as the gravestones in the film [39]. Furthermore, a multiplicity of perspectives that do not necessarily privilege that of the human researcher is acknowledged in new materialist thought such as Karen Barad's diffractive methodology (2007) [40].

During a visit to local café, the researchers noticed that a customer and two members of staff could be identified; luckily, they were quickly able to gain their consent during the filming. When editing the film, Aristotle's notion of luck (Tyche)—circumstances that were out of the researchers' control—came into play [11,20]. When filming in a crowd it was hoped that individuals would not be identified, however this proved not to be the case. This posed a dilemma for the researchers who had not wished to include people without their consent. The scenes could not be cut as they contained important pieces of discussion, and there were not resources to reshoot the project. If these people could be tracked down then retrospective consent could be gained, but this would be time consuming and perhaps impossible to do. The ethics policy was silent on filming people in crowd scenes so an alternative guide from Channel Four (2020) was examined [36]. At this point the interplay of the universal (policies) and particulars (filming by researchers in the local streets) was apparent. It was decided that the faces would be blurred out even though this would be detrimental to the film's aesthetics. Here, there was also a tension between two competing ethical values. Being creative practitioners, it can be seen that part of research integrity is to create work that is authentic and made with aesthetic rigour. At the same time, it was important to protect people's identities, and the researchers risked causing upset if they did not do this. There would be undesired consequences to either course of action, however, the researchers felt they had made the right decision. Who needed blurring out, and who was not featured/visible enough to need blurring out also resulted in significant ethical discussion amongst the production team. This was an example of needing to take each circumstance on a case-by-case basis and to sometimes consider other perspectives than our own to do what feels right in a particular context, even if it is above and beyond industry standard. Additionally, in terms of film production, the decision to use royalty-free music in order to comply with copyright laws and to give appropriate credits at the end of the

film in recognition of contributions, were examples of the team operating within standard ethical code for the film and television industries.

### *6.3. The Third Orientation*

The third orientation was the ethical deliberation about the film from the audience's perspective. This proved to be very useful and showed the 'blind spots' of the researchers. Initially, a 15 min film was thought to be an accessible way of encouraging staff and students to talk and think about ethics. However, the needs of people with hearing impairments or people from our international communities were not fully considered. The audience from the Ethics Sub-Committee saw this immediately. The researchers rectified this, and felt the film was enhanced by the addition of subtitles. The feedback led to further reflection from the researchers, who subsequently thought that a series of films would be needed that reflected the ethical positions from the University's diverse constituencies. A bigger question was the Eurocentric nature of the ethical frameworks in which many universities operate. The 'perceptiveness', as understood by Aristotle as an aspect of practical wisdom, of the researchers to see that issues needed ethical consideration, was sometimes lacking.

### 7. Conclusions

The original aim of the film was to provide a simple narrative based on two protagonists deliberating about ethics as they walked around their local environment, emulating what they would do every day. It was created in the spirit of Nussbaum's (2001) claim that stories can help people develop a moral sensibility and imagination [11]. However, *A Conversation about Ethics* remains an incomplete project for two main reasons. Firstly, the value of the project seems to lie in the deliberations that happened through making the film, rather than the ethical discussions it represented. Aristotle's assertion that deliberation needs to be practiced has been illustrated by this project. It also confirms that a practice-based approach to the research was very appropriate. The implications for art and design education are that ethics needs to integrated into all briefs and assignments. However, there are no 'tool kits' for making ethical judgements—art students and emerging researchers still need to be attuned to raising those questions and noticing how power can operate. Learning opportunities should be sought where staff and students can deliberate together, as part of the accepted business of the art school.

Tensions that exist between phronesis (ethical deliberation) and poetics (aesthetics and artistry) were also highlighted by this project and may be one of the reasons why art education has previously shied away from foregrounding ethics in the curriculum. There need not be a tension—Ricoeur (1994) and Scott (2021) have written about the poetic possibilities thrown up by phronesis [41,42]. By thinking deeply about the needs of others, creative practitioners can become more innovative, not constricted. These debates will enhance the art and design curriculum for students.

The second reason why the project is incomplete is that it remains ethically flawed, in spite of the best intentions of the researchers. It did not fully embrace the values of inclusion and diversity as a very ethnocentric perspective of ethical debate was represented. There were restraints on time and resources in making the film, and it could be argued that much was achieved through the hard work and professionalism of the researchers alongside their busy teaching commitments. However, the project team acknowledge that *A Conversation about Ethics* is really a work in progress. More people with different points of view need to join the discussion and carry the project forward.

The role of the ethics policy and also the policies from relevant organisations were important in helping the researchers make some decisions. They gave a framework for ordering the researchers' thinking. However, this too, is in some sense, an incomplete project. As new situations arise, policies need to be ever-evolving documents that encapsulate the continuing good deliberations of creative practitioners.

**Author Contributions:** Conceptualization, S.B. and K.T.-G. and S.H.; methodology, S.B. and K.T.-G.; software, S.H.; validation, S.B. and K.T.-G. and S.H.; formal analysis, S.B. and K.T.-G. and S.H.; data curation, S.H.; writing—original draft preparation, S.B.; writing—review and editing, S.H. and S.B.; visualization, S.H.; project administration, S.B. All authors have read and agreed to the published version of the manuscript.

**Funding:** This research received no external funding.

**Institutional Review Board Statement:** The study was conducted in accordance with the Declaration of Helsinki, and approved by, Ethic Committee Name: Leeds Arts University Ethics-Sub-Committee (approval code: 19-06-19, no. 7 p. 73. and approval date:19 June 19).

**Informed Consent Statement:** Informed consent was obtained from all subjects involved in the study.

**Data Availability Statement:** Not applicable.

**Conflicts of Interest:** The authors declare no conflict of interest.

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
