# Peer review of "A Conversation about Ethics: A Deliberative and Practice-Based Approach to Ethics in Arts Education"

_societies, doi:10.3390/soc13020039_

Round 1
Reviewer 1 Report
Dear authors,
thanks so much for your research. I was very excited to see a practice-based project that dealt with ethics heads-on.
I really appreciated how the paper introduced and defined certain key concepts (phronesis, poetics, fragility of goodness, tyche, etc). I am certain having these contextualised in practice-based research is useful for the field.
Here are some suggestions I hope you find useful:
- I would have been interested in seeing a more thorough state of the arts when it comes to ethics and artistic, practice-based research (and the institutions that host this type of research). I am sure that other efforts in having this conversation have taken place, instead of dismissing it as something potentially "alien and innapropiate" (line 23). For instance, the HKU (to which I bear no relation whatsoever) had a research line that dealt with this: https://www.hku.nl/en/research/projects/falk-hubner-common-ground.I am sure there are many others.
- I found the balance between storytelling and content hard to wrap my head around (this is very much a subjective take, of course). I craved more detailed discussions on some of the ethical concepts introduced, possibly using examples; and then I found that most of the stories were written under methodological issues. I appreciate how these were important in the project, and to practitioners reading the paper, but sometimes it felt like those could be a bit more condensed (again, personal preference!). I may have preferred more storytelling in the discussion instead, or even in a section that brought theory to practice (which I am aware is what the film attempts, so maybe this is way beyond what the paper intends to do). But since "the value of narrative" is, again, central here (abstract, line 11), I thought I'd still mention it.
Some typos and snafus you may want to revise:
There are weird spacings in the text (like in line 35). I think the text that begins at line 122 is a block quote that is not formatted as such. In line 170 "sort" may be "sought". In the paragraph that starts at line 222, there are numbers inserted in the text that do not belong there (possibly line numbers).
Thanks again for all your work, I hope some of the above can be of use.
Author Response
Thank you very much for your feedback and advise, based on a thoughtful reading of our work. We are grateful that you think this article will be relevant to the readers of Societies. Your comments have been very helpful.
We have noted the issues with the formatting and spacing and have tried to rectify these.
We agree that some contextualisation of ethics within practice or arts-based research is required in the introduction so we have added a section that discusses a range of research including the source you kindly shared.
We appreciate the comments about theory and practice and have integrated some additional theoretical points in the narrative, such as a note on humanist critique and a note about the gaze of the camera.
Reviewer 2 Report
Dear Author,
This is a well-crafted and engaging paper. I have truly enjoyed watching the 15 minutes video A Conversation about Ethics. The article offers an interesting approach to creating a resource for students about some ethical issues in qualitative research. I appreciated the reference to Martha Nussbaum's work and the astute take on Aristotle's concept of phronesis or practical wisdom to ground the pedagogical strategy of creating a walking video and conversation between two academics. The discussion section brought forward points which are relevant and interesting to read, while not particularly new - perhaps because I work within a university with strict protocols even for arts-based researchers? Overall, for the readers of Societies, the article will certainly inform practitioners, students and be a rich thinking space for further development of such videos; the walking conversation remains a terrific pedagogical idea to creatively and meaningfully bring to life a topic that might seem 'dry' and uninteresting to many students.
Here is some of my thinking along:
Line 23. "Academic from creative subject areas need to make sense of ideas that may at first seem alien and inappropriate" I suggest nuancing. Indeed there is a concern about the space of research and that of the arts. On that topic I suggest Donal O'Donoghue (2009) Are we asking the wrong questions in arts-based research? And yet arts-based researchers are deeply concerned about questions of ethics in research: the way power circulates, who/when not to represent, issues of appropriation, etc. I wonder why the paper did not at least mention some key studies in arts-based research such as the Barone & Eisner (2011), or Susan Finley et al. (2020) Arts-Based research.
Line 143. I appreciated this summary "the theoretical approach that underpins this project is based on an Aristotelian understanding that ethics are concerned with the universal and the particular." This is what I loved about the video; while the two researchers talk about larger ethical questions, it is the work of the camera that becomes the 'eye' of the particular. This eye of the camera points to the practical ethical questions in research - as you mention in your discussion, the need to ask for release forms or hide the identity of people. It would have been interesting to engage further on 'what does the camera do' in this context. The camera is the gaze, also noticing textures of place, almost as a metaphor of the complexities in-dwelling that space between the universal/particular.
Line 152. I cannot help but think about the important work of Sarah Pink on video ethnography. I highly suggest Pink (2007). Walking with video published in Visual Studies. While I understand that the goal of the video was not ethnographic (but rather pedagogical), Pink takes care to address the impact of the research tool itself, the camera. In what way are our methods raising ethical questions in arts-based/creative research?
Line 291. On the need to decenter the 'self' of the researchers. This is an important question and much talked about topic and I suggest adding current references in light of the onto-epistemological critique of Western humanist research, the need to decolonize research, or see for in instance the new feminist materialism approaches to research.
Line 342. typo "The implications for art and design education are that ethics needs to be integrated into all briefs and assignments." Perhaps an obvious statement? As your article points out, there are no recipes to making ethical judgements but art students/emerging researchers still need to be attuned in raising those questions and noticing how power can circulate.
I certainly hope that you and your team will continue this project Conversation about Ethics. Thank you for the opportunity to think and learn.
With best regards in your scholarship and future projects.
Author Response
Thank you for looking at our article and we are very pleased that you also looked at the film. Your comments and observations have been very helpful, particularly the points about the gaze of the camera. We understood this intuitively but it was a 'light-bulb' moment that you explicitly pointed this out. This point will influence our thinking in future work and we have alluded to in this article, so thank you for that .
We agree that there needs to be some wider contextualisation of arts-based practice in which to frame this project. So we have included some additional reference to other art-based projects that have developed their thinking about ethics.
The work of Sarah Pink is very relevant as it employs a similar method of capturing walking and talking through moving image, so we have included a reference to this work.
The point about posthuman thought again is well made and have added some references and brief explanation of these ideas at the point where the researchers contemplate the gravestones.
We have expanded the point about integrating ethics on all briefs by saying there are no simple solutions, but ethics needs to be considered and practiced within the teaching and learning context.
Round 2
Reviewer 1 Report
Dear authors,
thanks so much for your thoughtful additions, especially in the introduction.
I hope this paper, the video featured in it, and all future endeavours stemming from them are an absolute success.
Reviewer 2 Report
typo: Line 34 should read Barone and Eisner.
typo: Line 35 should read Finley